# The Development of Symbolic Expressions for Fire Detection with Symbolic Classifier Using Sensor Fusion Data

**DOI:** 10.3390/s23010169

**Published:** 2022-12-24

**Authors:** Nikola Anđelić, Sandi Baressi Šegota, Ivan Lorencin, Zlatan Car

**Affiliations:** Department of Automation and Electronics, Faculty of Engineering, University of Rijeka, 51000 Rijeka, Croatia

**Keywords:** genetic programming, symbolic classifier, fire-alarm, oversampling methods, undersampling methods

## Abstract

Fire is usually detected with fire detection systems that are used to sense one or more products resulting from the fire such as smoke, heat, infrared, ultraviolet light radiation, or gas. Smoke detectors are mostly used in residential areas while fire alarm systems (heat, smoke, flame, and fire gas detectors) are used in commercial, industrial and municipal areas. However, in addition to smoke, heat, infrared, ultraviolet light radiation, or gas, other parameters could indicate a fire, such as air temperature, air pressure, and humidity, among others. Collecting these parameters requires the development of a sensor fusion system. However, with such a system, it is necessary to develop a simple system based on artificial intelligence (AI) that will be able to detect fire with high accuracy using the information collected from the sensor fusion system. The novelty of this paper is to show the procedure of how a simple AI system can be created in form of symbolic expression obtained with a genetic programming symbolic classifier (GPSC) algorithm and can be used as an additional tool to detect fire with high classification accuracy. Since the investigation is based on an initially imbalanced and publicly available dataset (high number of samples classified as 1-Fire Alarm and small number of samples 0-No Fire Alarm), the idea is to implement various balancing methods such as random undersampling/oversampling, Near Miss-1, ADASYN, SMOTE, and Borderline SMOTE. The obtained balanced datasets were used in GPSC with random hyperparameter search combined with 5-fold cross-validation to obtain symbolic expressions that could detect fire with high classification accuracy. For this investigation, the random hyperparameter search method and 5-fold cross-validation had to be developed. Each obtained symbolic expression was evaluated on train and test datasets to obtain mean and standard deviation values of accuracy (ACC), area under the receiver operating characteristic curve (AUC), precision, recall, and F1-score. Based on the conducted investigation, the highest classification metric values were achieved in the case of the dataset balanced with SMOTE method. The obtained values of $\bar{ACC}\;\pm \;SD\left(ACC\right)$, $\bar{AUC}\;\pm \;SD\left(ACU\right)$, $\bar{Precision}\;\pm \;SD\left(Precision\right)$, $\bar{Recall}\pm SD\left(Recall\right)$, and $\bar{F1-score}\pm SD\left(F1-score\right)$ are equal to $0.998\;\pm \;4.79\times 10^{-5}$, $0.998\pm 4.79\times 10^{-5}$, $0.999\pm 5.32\times 10^{-5}$, $0.998\pm 4.26\times 10^{-5}$, and $0.998\pm 4.796\times 10^{-5}$, respectively. The symbolic expression using which best values of classification metrics were achieved is shown, and the final evaluation was performed on the original dataset.

## 1. Introduction

The detection of fire [1] is achieved using fire detectors that sense one or more products resulting from the fire such as smoke, heat, infrared or ultraviolet light radiation, and gas. These products can be detected using heat, smoke, flame, or fire gas detectors. In households, the detection of fire is usually achieved using cheap and stand-alone smoke detector devices, while in non-domestic buildings, fire detection is achieved using fire alarm systems.

The smoke detector is a stand-alone device that senses smoke, which in most cases is an indication of fire. There are two types of smoke detectors based on their working principle, i.e., photoelectric and ionizing.

A photoelectric/optical detector [2] contains a source of infrared/visible/ultraviolet light (incandescent light bulb or light emitting diode (LED)), a lens, and a photoelectric receiver. All these components are enclosed in a chamber through which air that could contain smoke flows. The smoke detector using optical beams/projected-beam is emitting a beam of infrared or ultraviolet light, which is received and processed by separate devices or reflected using the reflector. In the case of an optical beam, the light emitted by the light source passes through the air and reaches the photo-sensor. The light intensity will be reduced due to smoke, airborne dust, or other substances. The circuitry detects the light intensity, and if it drops below a specific threshold the alarm will be activated.

The ionization smoke detectors [3] use a tiny piece of radioactive material that is located between two electrically charged electrodes. The air that passes between two electrodes is then ionized and causes an electrical current between the electrodes. The smoke particles in the air are disrupting the current, which activates the alarm. A literature investigation showed that the majority of scientific papers available online regarding fire detection from smoke detection involve the implementation of various deep convolutional neural networks (DCNN) such as ResNet [4], Inception [5,6], YOLO [7], and others. In these papers, the image/video data types were used. There is a small number of papers in which only smoke detectors were used in fire detection in combination with artificial intelligence (AI) algorithms. Besides fire detection using only smoke detectors, there are numerous research papers in which fire detection systems are based on sensor fusion systems and AI.

### 1.1. Fire Detection Systems Based on Fire/Smoke Detectors and AI

The adaptive neuro-fuzzy system (ANFIS) was used in [8] for the identification of a true fire incident by including the change rate of smoke, the change rate of temperature, and humidity in the presence of fire. The data that was used in ANIFS was collected from sensors (flame detector, humidity, heat, smoke sensors, etc.) and prepared using Fuzzy Logic. The accuracy in fire detection using ANFIS is 100%. In [9], the guideline was developed and presented for choosing the most optimal sensor combinations for accurate residential fire detection. Besides the combination of the sensor, the implementation of the neural network and Naive Bayes classifier in smoke/fire detection was also investigated and the highest accuracy achieved was 97% in the case of the neural network and 100% in the case of Naive Bayes. The artificial neural network (ANN) was used in [10] to investigate if ANN could detect fire and smoke. The inputs of ANN were obtained from a sensor system consisting of smoke density, temperature, and carbon monoxide sensors, respectively. The highest detection accuracy achieved using ANN was 98.3%. The probabilistic neural network (PNN) was used in [11] to detect fire using burning smell. In this investigation, seven different materials were scorched in a vacuum oven at various temperature points and were pushed out using vacuum pumps where it was sniffed using an electronic nose. The results showed that PNN achieved a mean classification accuracy of 94.18%. The performance of the gas sensor array for the detection of smoldering and plastic fires while rejecting a set of nuisances was investigated in [12]. The fire and nuisance experiments were conducted in a fire room of 240 [m${}^{3}$]. The PLS-DA and support vector machines (SVM) were used to evaluate the performance of different multivariate calibration models for this dataset. The PLS-DA showed 100% specificity and 85% sensitivity, i.e., the system has difficulties in detecting plastic fires (signatures are close to nuisance scenarios).

### 1.2. Fire Detection Systems Based on Sensor-Fusion and AI

The sensor-fusion [13] can be described as the process of combining data obtained from sensors in such a way that the resulting information contains less uncertainty when compared to data obtained when the sensors are used individually. To implement sensor fusion, the equipment used in an experiment must contain two or more sensors. The following papers describe the utilization of multiple sensors, implementation of the sensor fusion process, and the application of AI methods in fire detection.

In [14], the mobile robot platform equipped with a multi-sensor system (smoke, flame, and temperature measurement sensors) was developed to detect fire. The robot was developed to track virtually prescribed paths with obstacle avoidance and motion planning to scan and detect fire sources with a sensor fusion system. The results of the conducted investigation showed that a sensor fusion system provides more reliable detection than a one sensor-based system with an accuracy of 92%.

The sensor fusion system developed in [15] consisted of temperature, smoke, and CO density sensors. The system for processing data consisted of a 3-layers data fusion structure, i.e., signal layer that consisted of sensors and part decision-making process, the characteristic layer that consisted of an Expert database pickup unit and neural network pickup unit, and the decision layer that consisted of a fuzzy inferent data fusion system. The results of this system showed the error using this system reaches the value of $10^{-4}$.

The sensor fusion system consisting of the temperature sensor, smoke, and CO density sensor was developed in [16] to detect fire. The data obtained from sensor fusion adjust the weights of three kinds of sensors, and by repeating iterations, generates the final decision (fire or not). The results of the investigation showed that the proposed method achieves high classification accuracy.

Fire detection using a gas sensor array with sensor fusion was developed and investigated in [17]. The system consisted of eight AMS MOX sensors, PID alpha sense sensors, NDIR CO${}_{2}$ alpha sense sensors, Electrochemical CO alpha sense sensors, and Humidity and Temperature sensors, respectively. The partial least squares discriminant analysis (PLS-DA) has been used for the detection of fire with data obtained from a gas sensor array. The results of the investigation showed 97% sensitivity in fire detection, although the system produced a significant rate of false alarms, i.e., 35%.

A sensor fusion system consisting of smoke, light, and temperature sensors was proposed in [18]. All measured data were interpreted using Arduino and wirelessly transferred to Raspberry PI for subsequent processing. The experiments were conducted day and night in three phases No-fire, On-fire, and Post-fire. The experimental results in the No-fire and On-fire phases during daytime achieved an accuracy of 98%, while during nighttime achieved an accuracy of 97%. In the case of all three phases, the achieved accuracy during daytime and nighttime are 97% and 98%, respectively.

The sensor fusion system with a BP neural network was utilized in [19] to develop a system for early indoor fire detection. The sensor fusion system consisted of sensors for the detection of temperature, smoke, and CO concentration, and the data was used for early fire detection with a BP neural network. The results of this investigation show that fire detection time is shortened by 32%.

In [20], the authors have developed three types of systems to create a smart home environment that can be used for intelligent fire detection/alarm and these are: a wearable motion sensing device mounted on residents’ wrists and its corresponding 3D gesture recognition algorithm for convenient automated household appliance control system, a wearable motion sensing device mounted on residents’ feet for indoor positioning algorithm, i.e., pedestrian navigation system for smart energy management and the multi-sensor circuit module with intelligent fire detection-alarm algorithm. In addition to the three types of systems, the intelligent monitoring interface was developed to provide information about temperatures, CO concentrations, communicative environmental alarms, household appliance status, human motion signals, gesture recognition, and indoor positioning, respectively. The data from the multi-sensor circuit module was processed using a probabilistic neural network to predict safe, warning, and dangerous conditions of the living room, bathroom, and kitchen, respectively. The results of the investigation showed that for home safety and fire detection, a classification accuracy of 98.81% was achieved.

Environmental issues can be a major challenge in achieving high measurement accuracy with wireless sensor networks (WSN), and the main reason for this is noise uncertainty. To overcome this problem, in [21] the artificial neural network (ANN) was applied for the received signal strength indicator based indoor target localization in WSN. In this investigation, the authors investigated the performance of 11 different ANN training functions and the results showed that all training functions show higher Average Localization Error and the system is more consistent in providing better location estimates.

An intelligent WSN was proposed in [22] for early forest fire detection. The proposed system is based on data mining and data fusion, backed by decades of research in forestry. In this research, an ant colony optimization algorithm was used for the creation of the multi-sensory base stations. The proposed method greatly improves the deployment of nodes in real life and with this system, higher accuracy in fire detection can be achieved.

Sensor fusion describes the approach of combining a multitude of perceptive sensor readings, such as cameras, radars, lidars, and other sensor types, intending to form a single model of the environment in which the aforementioned sensor array, commonly mounted to an automated device, currently resides in [23]. While commonly used in automated driving, such an approach of combining a multitude of inputs is growing in use in various scientific and engineering fields [24,25]. Many common issues need addressing when discussing the sensor fusion approaches, which have been a common topic of research in previous years. Such an issue is discussed by Singh et al. [26], who develop an IoT-enabled helmet that serves to safeguard the health of mine workers. This shows a common issue that is well addressed by the sensor fusion systems—the development of warning generation devices. These devices use the sensor fusion approach to achieve a robust system for warning of possible danger and healthcare risks. One of the most common issues is the assumption that data is “clean”—in other words, each time frame allows for the collection of all sensor readings as clean, noiseless data, without time shifts. As shown by Rahate et al. [27], this is rarely the case. The authors demonstrate the multi-task modality fusion approach’s application to develop models that may be robust even in cases where 90% of the sensor data is missing. Another issue that arises is the need for fast processing of different data types. Vakil et al. [28] propose FERNN, an approach based on the neural network that allows for the fusion of radio frequency and electro-optical sensor data. A similar issue is also discussed by Wu et al. [19], which demonstrates an application for an indoor fire early warning through the use of a back-propagation neural network. The developed system correctly identifies test fires, while improving the detection time by a large margin of 32%. A large number of sensor data present in the sensor fusion systems can also present itself as an issue, which is discussed by Mian et al. [29] for the case of bearing fault diagnosis. The authors propose an analytic approach that analyzes the collected data using the neighborhood component analysis (NCA) and relief algorithm (RA). Such data can then be processed using a simpler and faster support vector machine (SVM) based model, without significant loss of performance. The current state of the research in the area, as shown by the discussed papers, directly points towards the usability of AI-based and statistical analysis methods for the application in the area of sensor fusion.

### 1.3. Definition of Novelty, Research Hypotheses, and Scientific Contribution

As seen from the previous literature overview, the AI/ML methods (neural networks) that have been used showed promising results in terms of classification accuracy. However, training these models and further implementation require reasonably high computational resources. In other words, these models require a lot of storage space and a lot of computational power to process new data and generate the output. These AI/ML models are difficult to implement in micro-controllers that are used in multi-sensor systems since these micro-controllers are acquiring the data from multiple sensors and performing sensor-fusion processes. So, implementing these trained AI/ML models in fire detection systems would require some additional computational resources.

To overcome this problem, the novelty of this paper is to show the procedure of how using a simple GPSC algorithm the symbolic expression can be obtained that can detect fire with high classification accuracy. The obtained symbolic expression requires less storage space and can be easily integrated with a microcontroller to detect fire using data acquired from multiple sensors when compared to other AI/ML algorithms.

The GPSC is an algorithm that begins by creating a population of naive symbolic expressions that are unfit for a particular task and in each generation with the application of genetic operations to fit them for the particular task. The GPSC algorithm is an evolutionary algorithm; however, it has some similarities with supervised learning algorithms since it requires the dataset (input and output) to generate symbolic expressions.

To obtain the symbolic expression for fire detection, the GPSC algorithm will be applied to a publicly available dataset [30]. The dataset consists of data collected from sensors that measured aerosol concentration (number of particles and their size), humidity, temperature, air pressure, and gas. For this dataset, different scenarios were considered, and some of them are normal indoor, normal outdoor, indoor wood fire (firefighter training area), outdoor (wood, coal, and gas grill), and outdoor high humidity. A detailed description of the dataset is given in the following section. Based on the literature overview and described idea/novelty of this paper, the following questions arise:Is it possible to utilize the GPSC to obtain symbolic expression that could detect fire with high classification accuracy?Is it possible to balance the dataset class samples using different undersampling/ oversampling methods, and do balancing methods influence the classification accuracy of obtained symbolic expressions?Is it possible to achieve high calcification accuracy using a random hyperparameter search method for GPSC algorithm with 5-fold cross-validation?Is it possible to achieve high classification accuracy with the application of the best symbolic expression obtained with one of the balancing datasets on the original dataset?

The scientific contributions of this paper are:Investigate the possibility of GPSC application to the publicly available dataset for the detection of fire.Investigate if dataset balancing methods have any influence on classification accuracy of obtained symbolic expressions.Investigate if GPSC with random hyperparameter search method and 5-fold cross-validation can produce the symbolic expression with high classification accuracy in fire detection.Investigate if using the best symbolic expression can produce high classification accuracy in fire detection on the original dataset.

The outline of this paper is divided into sub-seeding sections, i.e.: Section 2—Materials and Methods, Section 3—Results and Discussion, and Section 4—Conclusions. In Section 2, the research methodology, dataset description, dataset balancing methods, genetic programming symbolic classifier, random hyperparameter search method, 5-fold cross-validation, evaluation metrics, methodology, and computational resources, are presented. In the Section 3, the results of using GPSC with random hyperparameter search method and 5-fold cross-validation are presented, as well as the final evaluation of the best symbolic expression on the original dataset. Also in Section 3, the obtained results are discussed. In the Section 4, the conclusions are given that are based on the discussion section and hypotheses defined here in this section.

## 2. Materials and Methods

In this section, the research methodology, dataset description, dataset balancing methods, GPSC, random hyperparameter search method, 5-fold cross-validation, evaluation metrics, methodology, and computational resources are described.

### 2.1. Research Methodology

The graphical representation of research conducted in this paper is shown in Figure 1.

As seen from Figure 1, the original dataset was balanced using different balancing methods and these are:Random undersampling,Random oversampling,Near miss-1,Adaptive Synthetic (ADASYN),Synthetic Minority Over-sampling (SMOTE),Borderline Synthetic Minority Over-sampling (Borderline SMOTE).

However, before the application of balancing methods, the dataset was scaled using the Standard Scaler method. After balancing the original dataset using different methods 6 different variations of the dataset were obtained. Each dataset was split into train/test datasets in a ratio of 70:30. The 70% of the dataset was used for training, i.e., in 5-fold cross-validation. Before 5-fold cross-validation, the random hyperparameters of GPSC were randomly selected. After the 5-fold cross-validation process, the evaluation metrics were applied of obtained symbolic expressions and if the evaluation metrics are high (>0.99) the process continued to the final stage where the GPSC was trained on 70% with the same hyperparameters as in the 5-fold cross-validation process. On the other hand, if the evaluation metrics values after 5-fold cross-validation were lower than 0.99, then the process would start from the beginning by randomly selecting hyperparameters of GPSC. When GPSC final training was completed the symbolic expression was obtained, and it was evaluated on the train and test dataset to obtain mean and standard deviation values of evaluation metrics. If the values of all metrics at this stage were above 0.99, the process was completed and if that was not the case the process would continue from the beginning by randomly selecting GPSC hyperparameters.

When the best symbolic expressions are obtained, they are compared in terms of evaluation metric values and the size of the symbolic expression. The symbolic expression that achieved the highest classification accuracy and size is smaller compared to other symbolic expressions and will be chosen as the best symbolic expression. The final evaluation of the best symbolic expression will be performed on the original dataset.

### 2.2. Dataset Description

As stated in this paper, a publicly available dataset was used, which can be downloaded from Kaggle [30]. The dataset was collected as part of the project in which a smoke detector that is based on artificial intelligence sensor fusion was used to determine if there is a fire or not. Sensor fusion can be described as a process of combining sensor data or data derived from disparate sources such that the resulting information has less uncertainty than the information collected when these sources were used individually. Sensors that were used to acquire this dataset were Sensirion SPS30 [31] (photoelectric smoke detector) for particulate matter and number concentration parameters, humidity sensor BME688 [32], temperature sensor SHT31 [33], air pressure BMP390 [34] and BMP388 [35], and gas sensors SPG30 [36] and BME688 [32]. The output of this sensor was measured in a particular matter (PM 1.0, PM 2.5) and number concentration (NC 0.5, NC 1.0, NC 2.5). All the other sensors were positioned around the SPS30 smoke detector. All data was collected using Nicla Sense ME board [37].

According to [30], the main idea of this project was to collect the data using different environmental information from sensor fusion to improve fire detection. There are different sensor fusion algorithms that could be implemented such as non-linear functions, thresholds, or linear regression. However, the author of this dataset chose linear regression-based sensor fusion since a lot of final output depends on the large number of sensor readings that have different correlations.

The data was collected in different environments and fire sources. The different scenarios used for the collection of data were considered, and these are:Normal indoor,Normal outdoor,Indoor wood fire,Indoor gas fire,Outdoor wood, coal, and gas grill,Outdoor high humidity.

The initial dataset consists of 60,000 readings without null values. The sample rate used to collect data is 1 [Hz] for all sensors. The initial dataset consisted of 15 variables and these are:Timestep UTC [s],Air temperature [C],Air humidity [%],Total volatile organic compounds (TVOC) [ppb],CO${}_{2}$ equivalent concentration [ppb],Raw H${}_{2}$—raw molecular hydrogen,Raw ethanol gas,Air pressure [hPa],Particulate matter size 1.0 μm (PM 1.0),Particulate matter size 2.5 μm (PM 2.5),Number of concentration of particulate matter (NC 1.0),Number of concentration of particulate matter (NC 2.5),Sample counter CNT,Fire alarm (fire alarm not activated—0, fire alarm activated—1).

However, UTC and CNT were omitted from further investigation since the idea was to develop symbolic expressions for fire detection using only sensor data as input variables. So, the dataset used in this paper consisted of 12 input variables and these are temperature, humidity, TVOC, eCO${}_{2}$, Raw H${}_{2}$, Raw Ethanol, Pressure, PM1.0, PM2.5, NC0.5, NC1.0, and NC2.5, while Fire Alarm labeled “target” was the output variable. All values of statistical analysis are given in Table 1.

To get a better insight into input variables when the fire alarm was triggered or not, the data was grouped into two categories, i.e., when the fire alarm was not activated and when the fire alarm was activated. The results are shown in Table 2 and Table 3.

As seen from Table 2 and Table 3, the mean temperature that does not cause a fire alarm is 19.69 [°C], while the mean temperature that causes fire alarming is equal to 14.48 [°C]. The mean humidity that does not cause the triggering of a fire alarm is 42.93%, while the mean humidity at which a fire alarm is triggered is equal to 50.78%. The mean total volatile organic compound at which the fire alarm is not triggered is equal to 4596.587 [bbp], while the mean total volatile organic compound at which the fire alarm is triggered is equal to 882 [bbp]. The mean value of CO${}_{2}$ equivalent concentration at which the fire alarm is not triggered is equal to 962.58 [ppm], while the mean value at which the fire alarm is triggered is equal to 553.19 [ppm]. The mean value of raw H${}_{2}$ at which the fire alarm is not triggered is equal to 12,896.31 [ppm], while the mean value at which the fire alarm is triggered is equal to 12,960.87 [ppm], respectively. The mean value of raw ethanol existence at which the fire alarm is not triggered is equal to 20,082.82 [ppm], while the mean value at which the fire alarm is triggered is equal to 19,623.05 [ppm]. The mean value of air pressure at which the fire alarm is not triggered is equal to 938.1 [hPa], while the mean value at which the fire alarm is triggered is equal to 938.8 [hPa]. For particulate matter 1.0, the mean value at which the fire alarm is not triggered is equal to 261.98 [ppm], while the mean value at which the fire alarm is triggered is equal to 36.14 [ppm]. In the case of 2.5, these values are 450.03 [ppm], and 78.4 [ppm], respectively. In the case of particulate matter concentrations, in the case of 0.5, 1.0, and 2.5, these values are 1356.35, 146.1, 493.8, 87.7, 178.98, and 40.5 [ppm], respectively. What is interesting to notice from Table 2 and Table 3 is the number of samples per class. In the case with no triggering of a fire alarm, the number of samples is equal to 17,873, while in the case of triggering a fire alarm the number of samples is equal to 44,757. Since the dataset is very imbalanced, previously mentioned balancing methods will be applied to equalize the number of samples between classes.

Generally, in Table 1, Table 2 and Table 3, the range of values (min-max) differ from variable to variable. Some of the input variables (temperature, humidity, and pressure) have very small ranges, while 9 of the 12 variables have very large ranges. Initial research with the GPSC algorithm showed that symbolic expressions with good classification accuracy (0.9–0.97) can be obtained with the original values, however, in order to increase the classification accuracy, the input variables were scaled using the Standard Scaler method.

Standard Scaler method [38] is a method used to standardize input variables by removing the mean and scaling unit variance to zero. The standard score of a dataset sample is calculated using the following expression:(1)$$z=\frac{x-u}{s},$$
where *u* is the mean of the input variable samples, *s* is the standard deviation of input variable samples, and *x* is the input variable sample. It should be noted that only input variables were scaled using Standard Scaler, and the target (output variable) was left in its original form.

To investigate the correlation between each input variable and the target (output variable), Pearson’s correlation analysis [39] was performed. The values of Pearson’s correlation analysis range from −1 to 1, and 0 is the worst possible correlation analysis between input and output variables. If the correlation value between the input and output variable is equal to −1, this means that if the value of the input variable increases the value of the output variable decreases and vice versa. On the other hand, if the correlation value between the input and output variable is equal to 1, this means that if the value of the input variable increases the value of the output variable also increases, and if the value of the input variable decreases the value of the output variable also decreases. The best possible ranges of correlation values are −1 to −0.5 and 0.5 to 1. The range from −0.5 to 0.5 represents a low correlation range, which can indicate potential difficulties in ML model development during the training process and in the end low classification accuracy of trained ML models. However, this is not always the case. The results of Pearson’s correlation analysis in the form of the heatmap are shown in Figure 2.

As seen from Figure 2, few variables, i.e., Humidity, Pressure, and Raw Ethanol, have some correlation to the target variable. The remaining nine variables have low correlation values to the target variable. It is interesting to note that all data collected from Sensirion SPS30 have between each other very high correlation values in the range of 0.63–0.99. However, in this investigation, all input variables will be used in GPSC to obtain symbolic expressions to see which input variables were included in the symbolic expressions with the highest classification accuracies. To better visualize the correlation between input variables and the output variable (target—fire/no fire alarm) the correlation values have been ranked from smallest to largest, as shown in Figure 3.

As previously stated and seen from Figure 3, three input variables, i.e., Raw Ethanol, Air Pressure, and Humidity, have the highest correlation with the Fire Alarm (target). The worst possible correlation is between particular matter variables and the fire alarm (target), which potentially indicates that these variables will not end up in the symbolic expression during GPSC execution.

### 2.3. Data Balancing Methods

Since the original dataset has a different number of samples per class, i.e., 17,873 (class 0—no fire) and 44,757 (class 1—fire), the original dataset is imbalanced. Since the imbalanced dataset can have a major negative impact on the performance of the supervised learning methods (ML algorithms) as reported in [40], the idea is to utilize under-sampling and over-sampling methods to balance the dataset by equalizing the number of samples per class. To clarify the terminology that will be used throughout this paper, the class with a smaller number of samples is called a minority class, while the class with a larger number of samples is called a majority class. In this paper, as previously stated, the following balancing methods were used, i.e., random undersampling, near miss-1, random oversampling, ADASYN, SMOTE, and Borderline SMOTE. These algorithms are briefly described in the following subsubsections.

#### 2.3.1. Undersampling Method: Random Undersampling

The random undersampling method [41] is a method of randomly selecting samples from the majority class to match the number of minority classes. After the number of samples of randomly selected numbers from the majority class is matched to the number of samples of the minority class, the process of dataset balancing is completed.

#### 2.3.2. Undersampling Method: Near Miss

The near miss method [42] is a name for a collection of undersampling methods that select samples based on the distance between majority and minority class samples. There are three different Near miss variations, i.e., Near miss 1, 2, and 3. The distance in all these methods is determined in feature space using Euclidean distance. In the NearMiss-1 method, the majority class samples with a minimum average distance to three close minority class samples. In NearMiss-2, the majority class samples with a minimum average distance to three further minority class samples. In NearMiss-3 the majority class samples with minimum distance to each minority class sample. In this paper, the NearMiss-1 method was used.

#### 2.3.3. Oversampling Method: Random Oversampling

The random oversampling method [43] increases the number the samples from the minority class by randomly selecting samples from the same class. The procedure is done until the number of samples from the minority class reaches the number of samples from the majority class. By matching the number of samples from the majority class, the dataset is balanced.

#### 2.3.4. Oversampling Method: ADASYN

The Adaptive Synthetic (ADASYN) method [44] starts with defining the majority and minority class by the number of samples. The first step is to calculate the degree of the class imbalance, which can be in the range between 0 and 1. If the value is very small, then the ADASYN algorithm begins its execution. The second step is to calculate the number of synthetic data samples of the minority class that have to be generated. For each minority class sample, find the K nearest neighbors based on Euclidean distance in n-dimensional space and calculate the ratio:(2)$$r_{i}=\frac{\Delta_{i}}{K}\;i=1,\ldots ,m_{s},$$
where $\Delta_{i}$ is the number of examples in K nearest neighbors around the minority class sample that belongs to the majority class, and $m_{s}$ is the number of minority class samples. The range of the ratio $r_{i}$ is between 0 and 1. The next step is to normalize the $r_{i}$ to obtain the density distribution. Then, calculate the number of synthetic data samples that have to be generated for each minority class. Finally, for each minority class sample, generate synthetic data samples, and the process is complete after the number of minority class samples matches the number of majority class samples.

#### 2.3.5. Oversampling Method: SMOTE

The synthetic minority oversampling technique (SMOTE) is a method of synthetically generating samples of the minority class to match the number of samples from the majority class. As described in [45], the procedure of generating synthetic data using SMOTE method consists of the following steps:Take the difference between the feature vector(sample) under consideration and its nearest neighbor,Multiply the difference by a random number in the 0–1 range, andAdds the result to the feature vector under consideration.

#### 2.3.6. Oversampling Method: Borderline SMOTE

The Borderline SMOTE method [46] oversamples the borderline minority examples. The term borderline is the border between two classes, in this case majority and minority classes. The first step is to find the borderline minority class samples. Then, synthetic samples are generated from them and added to the original training set.

The results of the application of various undersampling and oversampling methods on the original dataset are listed in Table 4.

### 2.4. Genetic Programming Symbolic Classifier

Genetic programming is a method of generating an initial population of randomly generated population members that are unfit for a particular task and adjusting them to solve the particular task with the use of genetic operators crossover and mutation. As stated, the program starts by creating the population of naive symbolic expressions by randomly selecting elements (functions, constants, and variables) from the so-called primitive set. It should be noted that in GP, the population members are represented as tree structures, which are very important due to the fact that the size of each symbolic expression is not only represented by its length, but also by its depth.

The method used to create the initial population is ramped half-and-half, in which half of the initial population is created with full and the rest with grow method. Another benefit of utilizing this method is that the depth of the entire population is defined in the range, i.e., 3 to 12. By using this method, a larger diversity is brought initially when compared to using just the full/grow method. The population size and tree depth size are in GPSC defined with hyperparameters population_size and init_depth. After the initial population is created the population members have to be evaluated. In GPSC this is done in the following way:Calculate the output using the values of the input variables from the dataset,Use the output as the argument of the Sigmoid function that can be written as:
(3)$$S\left(x\right)=\frac{1}{1+e^{-x}}$$Calculate the log loss of the sigmoid output and the real output (for each instance).

After the fitness value was obtained for each population member, the tournament selection method was applied. In this method, the population members are randomly selected from the population. Then, from the selected population members, randomly selected members are compared, and the one with the lowest value becomes a winner of the tournament selection. On the winners of tournament selection, genetic operations are performed, i.e., crossover and mutation. The size of tournament selection is defined with hyperparameter tournament_size.

In GPSC on tournament selection winners, four different genetic operations were performed, and these are crossover, subtree mutation, hoist mutation, and point mutation. For crossover, two tournament selection winners are required where on the first winner the random subtree is selected. Then, on the second tournament winner, a random subtree is selected and is used to replace the randomly selected subtree of the first tournament winner. By doing so, a new population member for the next generation is created. In the case of the remaining three genetic operations (subtree, hoist, and point mutation), only one winner for each genetic operation is required. In the subtree mutation, the random subtree is selected from the tournament winner. Then, by randomly selecting elements of the primitive set, the subtree is created, and it replaces the randomly selected subtree on the tournament winner to create a population member of the next generation. In the hoist mutation, the random subtree is selected on the tournament winner, and on this subtree, a second subtree is randomly selected. Then, the second tree replaces the originally selected subtree to create a member of the next generation. In case of point mutation on the tournament winner, the nodes are selected at random. The variables are replaced with other variables, constants with other constants, and functions with other functions from the primitive set. However, in the case of a function, the number of function arguments must be the same. The GPSC hyperparameter for aforementioned genetic operations are p_crossover, p_subtree_mutation, p_hoist_mutation, and p_point_mutation. The sum of all these genetic operations should be near 1 or equal to 1. If the sum is lower than 1, then some tournament selection winners will remain unchanged, i.e., they will enter the next generation without genetic operators being applied to them. The termination criteria are responsible for the termination of GPSC execution. If there are no termination criteria applied to the system, the GPSC would execute indefinitely. In the case of GPSC, two hyperparameters are responsible for terminating its execution, and these are generations, and stopping_criteria. The generations hyperparameter represents the maximum number of generations in the current GP execution, and if that number is reached, the GP algorithm is terminated. The stopping criteria are the minimum value for the fitness function, and if this value is reached by one of the population members, the GPSC is terminated. Since stopping criteria are usually defined near 0 or 0 this criterion is never met by the GPSC algorithm so the execution of the algorithm is terminated after reaching the maximum number of generations.

The constants in the GPSC execution are defined with the hyperparameter const_range. This hyperparameter contains a range of constants that are used in GPSC to randomly selects numbers and add them to the symbolic expression. The maximum number of samples (max_samples) is the maximum number of samples selected from the dataset used for training the population during execution. The parsimony coefficient is the last hyperparameter that is responsible for penalizing large population members. During the GPSC execution, it can happen that the size of the population members rapidly grows without any benefit in the fitness function value. This is called the bloat phenomenon, which can be prevented with the application of the parsimony coefficient. This coefficient will penalize large programs by making them less favorable for selection.

Before the development of the random hyperparameter search method, the initial investigation was done with GPSC to define ranges of hyperparameters. This initial tuning, i.e., range definition is especially important for the parsimony coefficient since the slight change in its value could greatly influence the GPSC evolution process. If the value is too small, the population members could rapidly grow in a couple of generations, which could result in memory overflow. If the value is too large, the GPSC will generate unevolved population members with usually small classification accuracy. The ranges of all hyperparameters used in this research are listed in Table 5.

### 2.5. Random Hyperparameter Search

In the previous subsection, the GPSC method was described as well as hyperparameters. Before each GPSC execution, the hyperparameters were randomly selected from a predefined range. The range of hyperparameters was defined using a trial-and-error procedure. In the GPSC script, the random hyperparameters method was defined in the form of a function that is called each time before GPSC execution. The list of hyperparameters is shown in Table 5.

### 2.6. 5-Fold Cross-Validation

The 5-fold cross-validation process has been chosen to generate a robust symbolic expression that can be used to detect fire with high classification accuracy. The classical approach of dividing the dataset on train/test without the use of 5-fold cross-validation can produce high classification accuracy on a train dataset; however, unseen data can result in poor classification accuracy. So, classic train/test process can in some cases cause over-fitting. Some examples of 5-fold cross-validation applications can be seen in [47,48,49]. To prove that over-fitting did not occur, the mean and standard deviation of evaluation metric values obtained on the train/test dataset were calculated. The high values of the standard deviation of any evaluation metric used in this paper could indicate a large difference between the evaluation metric values achieved on the train and test datasets, respectively. However, in this research, large standard deviation values of evaluation metrics used did not occur. The process of performing the 5-fold cross-validation of GPSC with a random hyperparameter search method can be summarized in the following steps:Select random hyperparameters of GPSC algorithm from their predefined range.Perform 5-fold cross-validation and obtained mean values of accuracy, area under the receiver operating characteristic curve, precision, recall, and F1-Score.Termination criteria 1st stage—if the values of previously mentioned metric values are greater than 0.99 then proceed to final analysis, otherwise, start from the beginning by randomly selecting new GPSC hyperparameters and performing GPSC 5-fold cross-validation analysis all over again.Final evaluation—if the termination criteria are satisfied the same parameters that were used for GPSC, 5-fold cross-validation is used in this final stage. The final evaluation consists of the final training of GPSC and final testing of obtained symbolic expression on a train and test dataset to obtain mean and standard deviation values of accuracy, area under the receiver characteristic operating curve, precision, recall, and f1-score values.Final evaluation 2nd stage—if the mean values of previously mentioned evaluation metrics are above 0.99, the process is completed, otherwise the process starts from the beginning.

The previously described procedure is shown in flowchart form in Figure 4.

### 2.7. Evaluation Metrics and Methodology

To evaluate the obtained symbolic expressions using GPSC in this paper, the accuracy, area under the receiver operating characteristics, precision, recall, and F1-scores were used. However, the values of these evaluation metrics are not only shown for the test dataset. The idea of this paper is to show mean values of evaluation metrics provided by the standard deviation error to see how the symbolic expression performs on train and test datasets, respectively. The metrics used in this research are shortly described in the Section 2.7.1, while detailed evaluation methodology is described in the Section 2.7.2.

#### 2.7.1. Evaluation Metrics

As already stated the evaluation metrics used in this paper are accuracy (ACC), area under the receiving operating characteristic curve (AUC), Precision, Recall, and F1-Score. To describe each evaluation metric first basic classification terminology must be introduced. In classification, the evaluation metrics are calculated from true positive (TP), false positive (FP), true negative (TN), and false negative (FN) [50]. When the ML model correctly predicts the positive class this outcome is labeled TP while the correct prediction of the negative class is labeled TN. In the case when ML model incorrectly predicts the positive class, the outcome is labeled as FP while the incorrect prediction of the negative class is labeled as FN. These four parameters define the confusion matrix and are basic elements used to calculate ACC, AUC, Precision, Recall, and F1-socre.

Accuracy, according to [51], can be described as a fraction of predictions the ML model made correctly and can be calculated using the expression:(4)$$ACC=\frac{TP+TN}{TP+TN+FP+FN}.$$

The AUC score [52] is a result of computing the area under the receiver operating characteristic (ROC) curve.

The precision [53] is an evaluation metric used to measure the positive samples that are correctly predicted from the total number of predictions in a positive class. Mathematically the precision is the ratio between TP and the sum of TP and FP and can be written in the following form:(5)$$Precision=\frac{TP}{TP+FP}.$$

The recall [53] can be described as the ability of the ML model to find all positive samples. The recall is a ratio between TP and the sum of TP and FN, and can be written in the following form:(6)$$Recall=\frac{TP}{TP+FN}.$$

The F1-score [54] is a harmonic mean of precision and recall. The contributions of precision and recall to the F1-score. The F1-score is calculated using an expression that can be written as:(7)$$F1-Score=2\frac{precision\cdot recall}{\left(precision+recall\right)}.$$

The range of AUC, ACC, precision, recall and F1-score are all in the 0 to 1 range where 1 is the best and 0 is the worst possible score.

#### 2.7.2. Evaluation Methodology

Before explaining the evaluation methodology procedure, let us summarize the training and testing process once again. The initial dataset was divided into the ratio of 70:30. On the training dataset, the 5-fold cross-validation is performed with randomly selected hyperparameters. If the 5-fold cross-validation process is passed, the next step is the 1st stage termination criteria. In the 1st stage termination criteria, the mean values of ACC, AUC, precision, recall, and F1-score are calculated, and if all values are greater than 0.99, the process progress to the final evaluation using GPSC. However, if the mean values of all evaluation metric values are lower than 0.99 then the process starts from the beginning by selecting random hyperparameters and performing the 5-fold cross-validation.

If the process progress to final evaluation the GPSC is trained using a trained dataset with the same hyperparameters as in the 5-fold CV. After training the symbolic expression is obtained the expression is evaluated on the train and test dataset to calculate the mean and standard deviation values of the aforementioned evaluation metrics. When these values are obtained, the 2nd stage of termination criteria is performed, i.e., if the mean values of evaluation metrics are greater than 0.99 and if the standard deviation values are lower than $10^{-3}$ then the process is completed. Otherwise, the process starts all over again from random hyperparameter selection.

The evaluation metrics were used after the application of the training and testing dataset on the final symbolic expression and mean and std values are obtained from the evaluation metrics. The procedure evaluating symbolic expressions can be divided into two steps, i.e.,

First step: during the 5-fold cross-validation on the train part of the dataset (70%) obtain evaluation metric values on the train and fold dataset packets to calculate the mean values of evaluation metrics that are used in termination criteria. If all mean evaluation metric values are greater than 0.97 after the 5-fold cross-validation process is completed, then the final training/testing is performed (step two). Otherwise, the hyperparameters are randomly selected and the 5-fold cross-validation process starts again.Second step: after mean values of evaluation metrics obtained during the 5-fold cross-validation process passed the termination criteria, the final training/ testing is performed. Training is performed using GPSC on 70% of the dataset, and during this step, the symbolic expression is obtained. After obtaining the symbolic expression, the evaluation metric values are obtained on the training dataset, and on the test dataset, i.e., train and test datasets are applied to the symbolic expression to evaluate its performance. Obtained values of evaluation metric on train/test dataset are used to calculate the mean and standard deviation values.

### 2.8. Computational Resources

In this paper, all investigations were conducted on a laptop with AMD Ryzen 5 Mobile 5500U 6-core (12 threads) processor, and 16 GB of DDR4 RAM. The Python programming language (version 3.9.1) was used to create all scripts. The original dataset was balanced using undersampling and oversampling functions from the imblearn library (version 0.9.1). The statistical analysis and correlation analysis was done using pandas library (version 1.0.5). The GPSC algorithm in these investigations was imported from gplearn library (version 0.4.1). The 5-fold cross-validation and random hyperparameter search method were developed from scratch. To visualize all the results and correlation heatmap the matplotlib library (version 3.4.3) was used.

## 3. Results and Discussion

In this section, the results from conducted investigation are presented. First, the results obtained using GPSC with a random hyperparameter search method and 5-fold cross-validation are presented in terms of evaluation metrics. Then, the best symbolic expression is shown, i.e., the symbolic expression using which the highest classification accuracy was achieved. The best symbolic expression is evaluated on the original dataset and the results of evaluation metrics are shown.

Initially, the GPSC was applied to the original dataset without a random hyperparameter search method and 5-fold cross-validation. This was done to investigate the range of hyperparameters that will be used later in the random hyperparameter search method. This step was required due to the parsimony coefficient parameter value, which is very sensitive and can greatly influence the evolution of the population during GPSC execution. A large value can choke the evolution process and produce the symbolic expression with low classification accuracy, while small values can create a bloat phenomenon. On the other hand, the initial investigation was also done to set an extremely low value of stopping criteria since the idea was to terminate the GPSC execution when a randomly chosen maximum number of generations was reached. This was done to enable GPSC to reach the lowest fitness value possible. The initial investigation was also necessary to investigate which genetic operation had a greater influence on the evolution process. It was found that higher values of the crossover coefficient have a greater contribution to lowering the value of the fitness function, obtaining symbolic expressions with high classification accuracy.

The initial statistical investigation of the dataset shown in Table 1 showed that input variables have a different range of values, i.e., some of them have a small range, while the majority of them have a really large value range. These value ranges greatly influenced the classification accuracies of obtained symbolic expressions since the highest mean values of evaluation metrics were around 0.97. To improve the classification accuracies, all input variables were scaled using the Standard Scaling method.

### 3.1. Results Achieved with GPSC Using Random Hyperparameter Search Method and 5-Fold Cross-Validation

Each dataset variation was used in GPSC with a random hyperparameter search method and 5-fold cross-validation. The combination of hyperparameters using the best symbolic expression, which was obtained in terms of classification accuracy on each dataset variation, is listed in Table 6.

From Table 6, it can be noticed that majority of best symbolic expressions for each dataset variation were obtained with a large population except in the case of random oversampling, where the value is near to lower boundary as it shown in Table 5. The crossover coefficient was dominating genetic operation in all cases which is obvious since the range of this hyperparameter was set to 0.95–1. The stopping criteria value was set to an extremely low value ($10^{-7}-10^{-6}$) to ensure that each GPSC execution terminates when the maximum number of generations is reached. The parsimony coefficient value was set to an extremely low value. Although the bloat phenomenon did not occur, some of the best obtained symbolic expressions are pretty large (Table 7). The mean values of evaluation metrics with standard deviation (error bars) are shown in Figure 5.

As seen from Figure 5, the best symbolic expression in terms of evaluation metric values was achieved with a dataset balanced with SMOTE and Borderline SMOTE method. However, these two symbolic expressions will be investigated, and the goal is to select the symbolic expression that has high classification accuracy with a smaller size of symbolic expression. Since the standard deviation values are small and are hardly visible from Figure 5, the standard deviation values are shown in Table 7, alongside mean values, the average CPU time required to obtain each symbolic expression, and the size of each symbolic expression.

As seen from Table 7, when symbolic expression obtained on SMOTE and Borderline SMOTE datasets are compared, both have high and similar classification accuracy. However, the symbolic expression obtained on SMOTE dataset is smaller in size (length and depth). Based on the size, the symbolic expression obtained in the case of SMOTE dataset is the best symbolic expression. The final evaluation of this symbolic expression on the original dataset is shown in the following subsection.

Regarding the average CPU time, the dataset size was one of the influences since with the use of undersampling methods, the dataset is much smaller so the execution was faster. In the case of random undersampling and near miss-1 dataset, each split in 5-fold cross-validation was executed for 60 [min], so in total for 5-fold cross-validation, 360 [min] was required. The final train/test lasted for an additional 60 [min], so in total average CPU time for GPSC with random hyperparameter search and 5-fold cross-validation on these two datasets is equal to 360 [min]. In the case of Random oversampling, ADASYN, SMOTE, and BorerlineSMOTE, the datasets were oversampled, i.e., much larger so GPSC training on each split in 5-fold cross-validation lasted for 100 [min]. Only for 5-fold cross-validation was 500 [min] required and add to that the additional 100 [min] for final training and evaluation. So, in total, 600 [min] average CPU time for oversampled datasets. Obtained results showed that all symbolic expressions obtained on each dataset variation achieved a classification accuracy higher than 0.97. The symbolic expression with the lowest classification accuracy was achieved in the case of Near Miss-1, Random oversampling, and Adasyn datasets. The highest classification accuracy was achieved in the case of SMOTE and Borderline SMOTE datasets. However, the size of the symbolic expression in the case of the Borderline SMOTE dataset (length/depth = 450/43 vs. 140/25) was crucial, so the smaller symbolic expression obtained on the SMOTE dataset was chosen as the best symbolic expression.

### 3.2. The Final Evaluation of the Best Symbolic Expression

As previously discussed, the best symbolic expression in terms of evaluation metric values and the size (length/depth) of symbolic expression was obtained in the case of the SMOTE dataset. The best symbolic expression can be written in the following form:(8)$$\begin{array}{ccc} y_{1} & = & 5.8\log\left(0.26\sqrt{X_{0}-17.1}\right)+0.63\sqrt{X_{1}-46.86}+0.86\log\left(0.098(X_{1}-46.86)\right)\\ & + & 0.86\log\left(0.43\log\left(\log\left(\log\left(9.6\times 10^{-5}\left(X_{2}-2788.37\right)\right)\right)\right)+\frac{0.048\sqrt{X_{5}-19849.2}}{X_{6}-938.471}\right)\\ & + & 2.\log(1.44\log(0.43\log(\log(1.44\log(0.036\sqrt{X_{5}-19849.2}\csc(1.44\\ & & \log(\left(2.3\log\left(1.44\log\left(0.43\log\left(0.43\log\left(\log\left(\log\left(9.6\times 10^{-5}\left(X_{2}-2788.37\right)\right)\right)\right)\right)\right)\right)\right)\\ & & (\log(0.43\log(0.43\log(0.43\log(0.43\log(0.43\log(\log(\log(9.6\times 10^{-5}(X_{2}\\ & - & 2788.37)))))))))))))))))+2.\csc(0.75\left(X_{6}-938.471\right))\\ & & \sqrt{\log(1.44\log\left(\frac{10380.5\log(0.75\left(X_{6}-938.471\right))}{X_{2}-2788.37}\right))}\\ & + & 2.\sqrt{\log(1.44\log\left(1.44\log\left(\log\left(\log\left(9.6\times 10^{-5}\left(X_{2}-2788.37\right)\right)\right)\right)\right))}\\ & & \csc\left(0.75\left(X_{6}-938.471\right)\right)+2.\log(1.44\log(\log(1.44\log(\log(\log(9.6\times 10^{-5}(X_{2}\\ & - & 2788.37)))))))+0.86\log(\log(1.44\log(1.44\log(\log(0.43\log(0.43\log(\log(\log(9.6\times 10^{-5}(X_{2}\\ & - & 2788.37))))))))))+0.86\log(\log(1.44\log(1.44\log(\log(1.44\log(0.43\log(0.43\log(\log(\\ & & \log\left(9.6\times 10^{-5}\left(X_{2}-2788.37\right)\right))))))))))+2.8\log(1.44\log(\log(1.44\log(0.43\\ & & \log(0.43\log(0.43\log(0.43\log(0.43\log(0.43\log(\log(\log(9.6\times 10^{-5}(X_{2}\\ & - & 2788.37)))))))))))))+2.8\log\left(\log\left(0.75\left(X_{6}-938.471\right)\right)-0.003\left(X_{4}-12926.3\right)\right)\\ & + & 2.\sqrt{\log(1.44\log\left(0.003\left(X_{4}-12926.3\right)\right))}\csc(\sin(\sin(\sin(\sin(\sin(0.75(X_{6}\\ & - & 938.471)))))))+0.006\left(X_{4}-12926.3\right)\\ & + & 2.4\sqrt{\log(0.43\log\left(\log\left(\log\left(1.44\log\left(0.003\left(X_{4}-12926.3\right)\right)\right)\right)\right))}\\ & + & 0.86\log(\log\left(1.44\log\left(0.036\sqrt{X_{5}-19849.2}\csc\left(1.44\log\left(\frac{0.084\sqrt{X_{5}-19849.2}}{\log(\log\left(0.75\left(X_{6}-938.471\right)\right))}\right)\right)\right)\right))\\ & + & 0.036\sqrt{X_{5}-19849.2}+6.\log\left(1.44\log\left(\sin\left(0.75\left(X_{6}-938.471\right)\right)\right)\right)\\ & + & 2.4\sqrt{\log(0.43\log\left(\log\left(0.43\log\left(\log\left(1.44\log\left(0.43\log\left(\log\left(\sin\left(0.75\left(X_{6}-938.471\right)\right)\right)\right)\right)\right)\right)\right)\right))}\\ & & \csc(\sin\left(\sin\left(\sin\left(0.75\left(X_{6}-938.471\right)\right)\right)\right)). \end{array}$$

The best symbolic expression (Equation (Equation 8)) consists of the following input variables: temperature ($X_{0}$), humidity ($X_{1}$), TVOC ($X_{2}$), Raw H${}_{2}$ ($X_{4}$), Raw Ethanol $X_{5}$, and air pressure ($X_{6}$). This means that particular matter variables (PM1.0, PM2.5, NC0.5, NC1.0, and NC2.5) did not end up in the best symbolic expression. This is logical, since these variables have a high correlation with each other (0.64 to 1), but a very weak correlation with the target variable (fire alarm/no fire alarm) as seen from Figure 2.

The final evaluation of symbolic expression was achieved by applying this expression to the entire original dataset and measuring ACC, AUC, Precision, Recall, and F1-score, respectively. To evaluate the symbolic expression and calculate evaluation metrics, the following calculation procedure is required:The values of input variables of the original dataset are used to calculate the output of the symbolic expression,The output values of symbolic expression is used as input in Sigmoid function (Equation (Equation 3)) to calculate the output,The output of the Sigmoid function is then transformed to an integer value to obtain a 0 or 1 value.

It should be noted that the standard scaling method does not have to be applied to the original dataset since the symbolic expression can work with the original data. In other words the standard scaling formula (Equation (Equation 1)) is already included for each variable in Equation (Equation 8). The results are shown in Table 8.

As seen from Table 8, the results of ACC, AUC, Precision, Recall, and F1-score are slightly lower than evaluation metric values obtained on the SMOTE dataset shown in Table 7.

### 3.3. Results Summary

In this subsection, the obtained results are compared to the results from the literature described in the Introduction section. At the end of this subsection, the key observations on conducted investigations are provided.

The comparison of the results obtained in this investigation on the original dataset and previous investigation discussed in the Introduction is listed in Table 9.

As seen from Table 9, the results (classification accuracy) from other research papers in fire/smoke detection are in the 92–100% range. So, the results obtained in this paper are higher than the majority of results from other research papers. However, the research papers [8,9] in which the accuracy is 100% were achieved with ANFIS and ANN ML algorithms, which require more computational resources when compared to the symbolic expression obtained in this paper. However, in those papers, the authors have used balanced datasets to train their algorithms, which is an initial advantage when compared to the dataset used in this paper. Although, dataset balancing methods provide a good starting point for training ML algorithms, it is always better to obtain the original balanced dataset. Based on conducted investigation, some of the key observations are:The GPSC can be used to obtain symbolic expression which can be used to detect fire using sensory data obtained from the sensor fusion system. However, to achieve high classification accuracy tuning of GPSC hyperparameters is mandatory and was achieved with a random hyperparameter search method.The random hyperparameter search method is a good method for obtaining the combination of GPSC hyperparameters using which the highest classification accuracy of obtained symbolic expression can be achieved. However, due to the dataset size, computational resources used, and required average CPU time, this method is slow, but generates good results.Since the dataset was greatly imbalanced, i.e., a large number of samples in one class and a small number of samples in another class, the original dataset could not be used in the investigation and dataset balancing methods were applied. The application of the balancing method created a great starting point for analysis and in the end, the symbolic expressions with high classification accuracy were obtained.The best symbolic expressions were obtained on datasets balanced with SMOTE and Borderline SMOTE methods. However, in the case of the Borderline SMOTE method, the symbolic expression is three times larger in terms of length than in the case of SMOTE. So, the best symbolic expression based on size and accuracy was obtained in the case of SMOTE.The final evaluation of the best symbolic expression on the original dataset showed that this procedure is the procedure of handling imbalanced datasets, i.e., balance the dataset using different balancing methods, using them to train the ML algorithm and perform a final evaluation on the original imbalanced dataset.

## 4. Conclusions

In this paper, the GPSC with random hyperparameter search method and 5-fold cross-validation was applied to publicly available datasets to obtain robust symbolic expressions that could detect fire with high classification accuracy.

The conducted investigation showed that GPSC can generate symbolic expressions, which can be used for fire detection with high classification accuracy. The combination of GPSC with random hyperparameter search method and 5-fold cross-validation generated robust symbolic expressions for fire detection with high classification accuracy. The dataset balancing methods ADASYN, SMOTE, and Borderline SMOTE balanced the original dataset and provided a good starting point for the application of the GPSC algorithm. Using these dataset variations to train GPSC produced symbolic expressions with classification accuracy in the range of 0.97 to 0.999. So, the dataset balancing methods have an influence on the classification performance of obtained symbolic expressions. The applied procedure also showed that the best symbolic expression in terms of classification accuracy obtained on a balanced dataset can achieve almost the same classification accuracy when applied to the original (imbalanced) dataset. The investigation also showed that the best symbolic expression does not contain particular matter variables only temperature, humidity, TVOC, Raw H${}_{2}$, Raw Ethanol, and air pressure.

The advantages of the proposed method are:After application of GPSC, the symbolic expression is obtained that can be easily used regardless of its size since it requires lower computational resources to produce the solution when compared to other ML algorithms,The dataset balancing methods created a good starting point for the implementation of GPSC and using GPSC symbolic expressions with high classification performance were obtained,The GPSC with random hyperparameter search method and 5-fold cross-validation generated the symbolic expressions that are robust and have high classification accuracy.

The disadvantages of the proposed method are:To implement a random hyperparameter search method, the ranges of each hyperparameter have to be defined by initial testing of GPSC. The population size, maximum number of generations, tournament size, and parsimony coefficient have a great influence on the execution time. The bigger the size of the population, tournament size, and larger maximum number of generations, the longer it will take the GPSC to execute. However, the parsimony coefficient is the most sensitive GPSC hyperparameter, and the range should be carefully defined. If the value of the parsimony coefficient is too small, it can result in a bloat phenomenon, while a very large value can prevent the growth of the symbolic expression, which will result in a small symbolic expression with poor classification performance.The dataset oversampling methods greatly influence the GPSC execution time since the dataset used to obtain symbolic expression using GPSC is much larger than the original one.

This approach showed how using a simple GPSC algorithm and the data obtained from the sensor fusion system a robust symbolic expression can be obtained that can detect fire with high classification accuracy. The symbolic expression can be potentially integrated into the micro-controller system, which controls the entire multisensor system to provide additional information if the fire is actually occurring in a multi-sensor environment. The other benefit of using symbolic expression when compared to other ML models is that this expression requires less computational power and memory than ML models, so it can be easily integrated with microcontroller devices such as Arduino.

The future work will be focused on development of multi-sensor system and collecting data to obtain a balanced dataset. After a reasonably large and balanced dataset is collected, other ML methods will be utilized to obtain ML models, which could detect fire with high classification accuracy. After different models are trained, they will be implemented on micro-controllers to measure the time required to detect fire by conducting different fire scenarios.

## Figures and Tables

**Figure 1 sensors-23-00169-f001:**
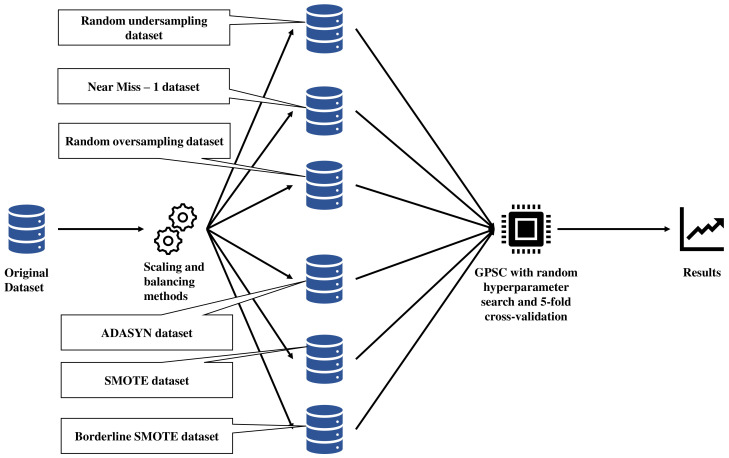
The schematic view of research methodology.

**Figure 2 sensors-23-00169-f002:**
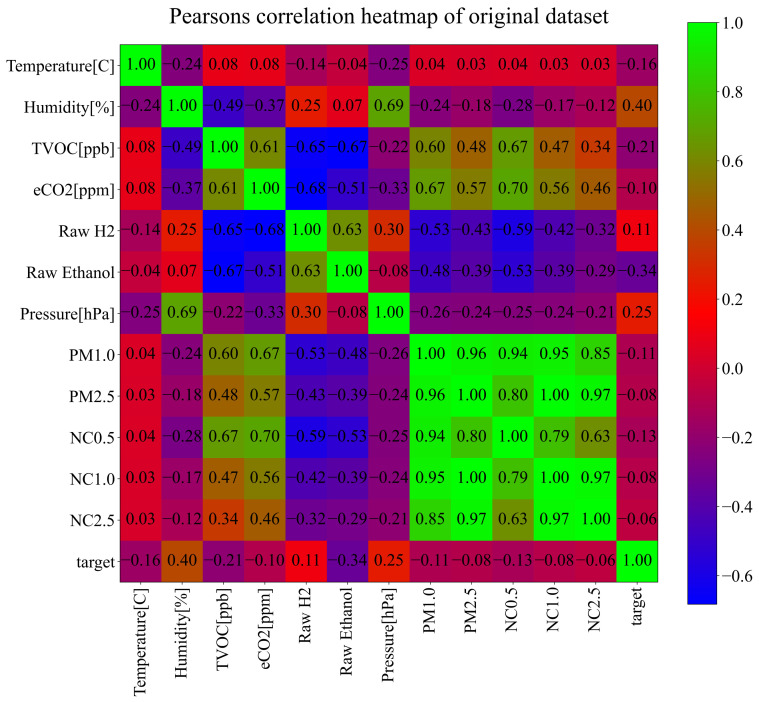
The results of Pearson correlation analysis performed on the original dataset.

**Figure 3 sensors-23-00169-f003:**
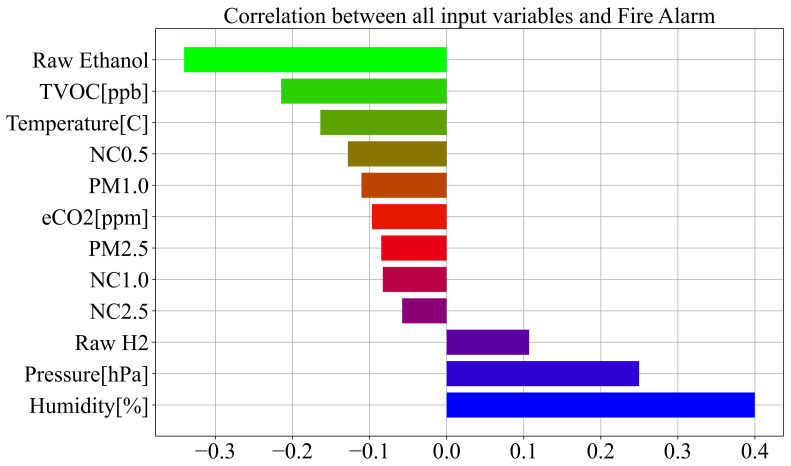
Correlation of all input variables with target (Fire Alarm) variable.

**Figure 4 sensors-23-00169-f004:**
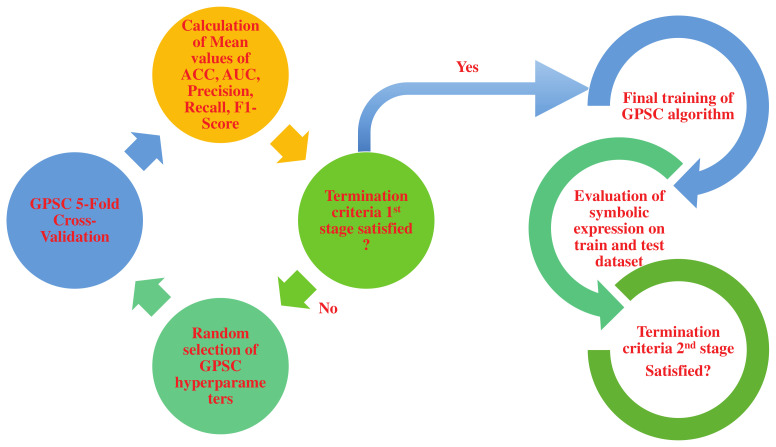
The schematic view of GPSC with random hyperparameter search and 5-fold cross-validation.

**Figure 5 sensors-23-00169-f005:**
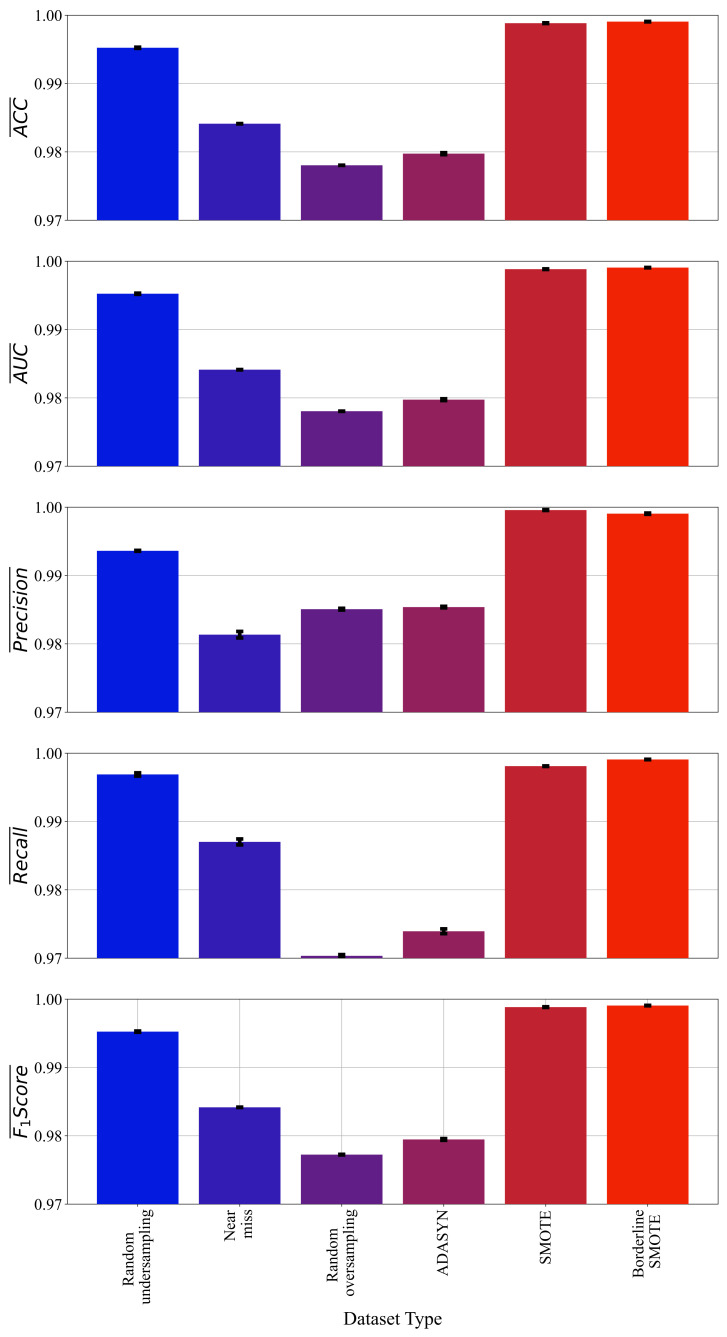
The mean and standard deviation of ACC, AUC, Precision, Recall, and F1-score values achieved with GPSC, random hyperparameter search, and 5-fold cross-validation on each dataset variation. (The standard deviation is shown in the form of error bars).

**Table 1 sensors-23-00169-t001:** The initial statistical analysis of the original dataset with variable names used in the GPSC algorithm.

Variable Name	Count	Mean	Std	Min	Max	GPSC Variable Representation
Temperature [C]	62,630	15.97042	14.35958	−22.01	59.93	$X_{0}$
Humidity [%]		48.5395	8.865367	10.74	75.2	$X_{1}$
TVOC [ppl		1942.058	7811.589	0	60,000	$X_{2}$
eCO${}_{2}$ [ppm]		670.021	1905.885	400	60,000	$X_{3}$
Raw H${}_{2}$		12,942.45	272.4643	10,668	13,803	$X_{4}$
Raw Ethanol		19,754.26	609.5132	15,317	21,410	$X_{5}$
Pressure [hPa]		938.6276	1.331344	930.852	939.861	$X_{6}$
PM1.0		100.5943	922.5242	0	14,333.69	$X_{7}$
PM2.5		184.4678	1976.306	0	45,432.26	$X_{8}$
NC0.5		491.4636	4265.661	0	61,482.03	$X_{9}$
NC1.0		203.5865	2214.739	0	51,914.68	$X_{10}$
NC2.5		80.04904	1083.383	0	30,026.44	$X_{11}$
target		0.714626	0.451596	0	1	*y*

**Table 2 sensors-23-00169-t002:** The statistical data of samples when fire alarm was not activated.

	Count	Mean	Std	Min	Max
Temperature [C]	17,873	19.6948031	14.9829319	−22.01	59.93
Humidity [%]		42.9300767	11.9628544	10.74	75.2
TVOC [ppb]		4596.58725	14,255.5756	0	60,000
eCO${}_{2}$ [ppm]		962.587255	2921.74993	400	39,185
Raw H${}_{2}$		12,896.3168	432.44162	10,668	13,803
Raw Ethanol		20,082.8235	956.339624	15,317	21,410
Pressure [hPa]		938.101383	1.23795718	931.131	939.861
PM1.0		261.982706	1439.7256	0	13,346.69
PM2.5		450.034639	2828.77478	0	41,262.98
NC0.5		1356.28382	7155.12266	0	61,482.03
NC1.0		493.872027	3150.39016	0	47,089.598
NC2.5		178.982259	1446.59579	0	26,916.836

**Table 3 sensors-23-00169-t003:** The statistical data of samples when the fire alarm was activated.

	Count	Mean	Std	Min	Max
Temperature [C]	44,757	14.4831516	13.8255854	−22.01	41.41
Humidity [%]		50.7795337	5.93723882	13.36	70.28
TVOC [ppb]		882.013071	548.606072	0	18,062
eCO${}_{2}$ [ppm]		553.189356	1275.26098	400	60,000
Raw H${}_{2}$		12,960.8781	167.385665	10,939	13,637
Raw Ethanol		19,623.0504	307.123385	17,809	21,109
Pressure [hPa]		938.837806	1.3090303	930.852	939.771
PM1.0		36.1464057	590.458583	0.15	14,333.69
PM2.5		78.4178419	1493.57607	0.16	45,432.26
NC0.5		146.111337	2144.94205	1.06	60,442.71
NC1.0		87.6655491	1689.24266	0.165	51,914.68
NC2.5		40.5416272	895.171419	0.004	30,026.438

**Table 4 sensors-23-00169-t004:** The number of samples per class after application of various undersampling and oversampling methods on the original dataset.

Dataset Balancing Method	Class 0	Class 1	Total
Random undersampling	17,873	17,873	35,746
Near Miss-1	17,873	17,873	35,746
Random Oversampling	44,757	44,757	89,514
SMOTE	44757	44,757	89,514
Borderline SMOTE	44,757	44,757	89,514
ADASYN	44,759	44,757	89,516

**Table 5 sensors-23-00169-t005:** The range of GPSC hyperparameters which were randomly selected in each execution of GPSC.

Hyperparameter Name	Lower Bound	Upper Bound
Population_size	500	2000
number_of_generations	200	300
tournament_size	100	500
init_depth	(3,7)	(7,12)
crossover	0.95	1
subtree_mutation	0.001	0.1
point_mutation	0.001	0.1
hoist_mutation	0.001	0.1
stopping_criteria	$1\times 10^{-7}$	$1\times 10^{-6}$
max_samples	0.99	1
constant_range	−100,000	100,000
parsimony_coeff	$1\times 10^{-5}$	$1\times 10^{-4}$

**Table 6 sensors-23-00169-t006:** The randomly chosen hyperparameters with which the symbolic expressions with the highest classification accuracy were obtained.

Dataset Type	GPSC Hyperparameters (Population_Size, Number_of_Generations, Tournament_Size, Initial_Depth, Crossover, Subtree_Muation, Hoist_Mutation, point_Mutation, Stopping_Criteria, Max_Samples, Constant_Range, Parsimony_Coefficient)
**Random** **Undersampling**	1477, 221, 406, (5, 12), 0.96, 0.013, 0.013, 0.012, $7.82\times 10^{-7}$, 0.99, (−13467.47, 36155.63), $5.45\times 10^{-5}$
**Near** **Miss-1**	1422,173,290, (6, 10), 0.96, 0.0059, 0.0075, 0.021, $9\times 10^{-6}$, 0.99, (−46197.99, 30568.98), $2.76\times 10^{-5}$
**Random** **Oversampling**	654,250, 383, (7, 11), 0.96, 0.023, 0.013, 0.0019, $6.79\times 10^{-7}$, 0.99, (−76506.62, 63083.63), $5.5\times 10^{-5}$
**ADASYN**	952, 252, 290, (7, 8), 0.96, 0.0051, 0.0016, 0.024, $1.56\times 10^{-7}$, 0.99, (−47945.94, 94095.29), $3.34\times 10^{-5}$
**SMOTE**	1111,217,190, (6, 11), 0.97,0.005,0.01,0.0085, $8.12\times 10^{-7}$,0.99, (−12456.11, 25100.79), $9.7\times 10^{-5}$
**Borderline** **SMOTE**	1194,108,180, (5, 8), 0.95, 0.014, 0.011, 0.013, $5\times 10^{-6}$, 0.99, (−87036.2, 28148.73), $3.65\times 10^{-5}$

**Table 7 sensors-23-00169-t007:** The numerical values of mean ACC, AUC, Precision, Recall, and F1-Score with standard deviation.

Data Type	$\bar{\mathrm{ACC}}$ ±SD(ACC)	$\bar{\mathrm{AUC}}$ ±SD(AUC)	$\bar{\mathrm{Precision}}$ ±SD(Precision)	$\bar{\mathrm{Recall}}$ ±SD(Recall)	$\bar{F1-\mathrm{Score}}$ ±SD(F1-Score)	Average CPU Execution Time [min]	Length/Depth of Symbolic Expression
**Random** **Undersampling**	0.9952 $\pm 8.652\times 10^{-5}$	0.9952 $\pm 8.67\times 10^{-5}$	0.9936 $\pm 6.53\times 10^{-5}$	0.996 $\pm 2.39\times 10^{-4}$	0.995 $\pm 8.67\times 10^{-5}$	360	188/32
**Near** **Miss-1**	0.984 $\pm 2.68\times 10^{-5}$	0.984 $\pm 2.57\times 10^{-5}$	0.981 $\pm 4.8\times 10^{-4}$	0.987 $\pm 4.34\times 10^{-4}$	0.984 $\pm 2.57\times 10^{-5}$	360	460/51
**Random** **Oversampling**	0.978 $\pm 1.58\times 10^{-5}$	0.978 $\pm 1.98\times 10^{-5}$	0.985 $\pm 1.28\times 10^{-4}$	0.97 $\pm 1.82\times 10^{-4}$	0.977 $\pm 4.71\times 10^{-5}$	600	83/23
**ADASYN**	0.979 $\pm 1.4\times 10^{-4}$	0.979 $\pm 1.34\times 10^{-4}$	0.985 $\pm 1.11\times 10^{-4}$	0.973 $\pm 3.5\times 10^{-4}$	0.979 $\pm 1.26\times 10^{-4}$	600	727/38
**SMOTE**	0.998 $\pm 4.79\times 10^{-5}$	0.998 $\pm 4.79\times 10^{-5}$	0.999 $\pm 5.32\times 10^{-5}$	0.998 $\pm 4.26\times 10^{-5}$	0.998 $\pm 4.796\times 10^{-5}$	600	140/25
**Borderline** **SMOTE**	0.999 $\pm 2.92\times 10^{-5}$	0.999 $\pm 2.919\times 10^{-5}$	0.999 $\pm 9.15\times 10^{-5}$	0.999 $\pm 3.07\times 10^{-5}$	0.999 $\pm 3.042\times 10^{-5}$	600	450/43

**Table 8 sensors-23-00169-t008:** The ACC, AUC, Precision, Recall, and F1-score achieved with application of $y_{1}$ equation on the original dataset.

Evaluation Metric	Value
ACC	0.9984
AUC	0.9986
Precision	0.9997
Recall	0.998
F1-score	0.9988

**Table 9 sensors-23-00169-t009:** The comparison of the obtained results with previous investigations discussed in the Introduction section.

References	Method	Fire or Smoke	Results
[8]	ANFIS	Fire	ACC: 100%
[9]	ANN, Naive Bayes	Smoke/Fire	ACC: 97–100%
[10]	ANN	Fire/Smoke	ACC: 98.3%
[11]	PNN	Fire	ACC: 94.8%
[17]	PLS-DA	Fire	Sensitivity: 97%
[14]	PLS	Fire	ACC: 92%
[15]	Neural Network	Fire	classification error $10^{-4}$
[16]	BP Neural Network	Fire	ACC: 98%
[18]	Dempster-Shafer Theory	Fire	ACC: 97–98%
[20]	PNN	Fire	ACC: 98.81%
This investigation	GPSC	Fire	ACC: 99.84%

## Data Availability

Not applicable.

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
