# Peer review of "The Development of Symbolic Expressions for Fire Detection with Symbolic Classifier Using Sensor Fusion Data"

_sensors, 2022, doi:10.3390/s23010169_

Round 1

Reviewer 1 Report

Please see attached Review Report

Author Response

The authors of this paper want to thank the reviewer for his time and effort in providing constructive comments and suggestions which significantly improved the quality of this manuscript. The authors do hope that the paper in this form is acceptable for publishing. 

The answer to comments and suggestions made by the first reviewer is listed below. 

The authors have made good contribution to the domain of smoke detection, however some of the following comments, I feel, are required to be fixed. 

  • I believe that the readability of this paper can be improved by writing the proposed ideas in clear, simple and concise manner. For instance, 
  •  Please make a separate section on “Related Work”. Please try to elaborate pros and cons of the existing state of the art in smoke detection systems. And then mention how your system is novel and going to contribute the area.  

Answer: In the original version of the manuscript the authors have separated the introduction section into following subsections, and these are:

  • Subsection 1.1. “Fire detection systems based on smoke detectors and AI” 
  • Subsection 1.2. “Fire detection systems based on sensor-fusion and AI” 
  • Subsection 1.3. “Definition of Novelty, Research Hypotheses, and Scientific Contribution”

As seen from the original version of the manuscript the Related work has already been presented in subsections 1.1 and 1.2. This means that a literature overview has already been done in the original version of the manuscript. 

Regarding the pros and cons of existing state of the art in smoke/fire detection, the authors would like to draw the reviewer's attention to the first paragraph of subsection 1.3 in the original version of the manuscript. “Citing from the original manuscript version (subsection 1.3, first paragraph): “As seen from the previous literature overview the AI/ML methods (neural networks) that have been used showed promising results in terms of classification accuracy. However, training these models and further implementation require reasonably high computational resources. In other words, these models require a lot of storage space and a lot of computational power to process new data and generate the output. These AI/ML models are difficult to implement in micro-controllers that are used in multi-sensor systems since these micro-controllers are acquiring the data from multiple sensors and performing sensor-fusion processes. So to implement these trained AI/ML models in fire detection systems would require some additional computational resources.

Regarding the novelty in this paper this was also provided in the original version of the manuscript. The novelty is given in the second paragraph of subsection 1.3 just after the pros and cons of previous approached. Citing from the original version of the manuscript (second paragraph of subsection 1.3): “To overcome this problem the novelty of this paper is to show the procedure of how using a simple GPSC algorithm the symbolic expression can be obtained that can detect fire with high classification accuracy. The obtained symbolic expression requires less storage space and can be easily integrated with a microcontroller to detect fire using data acquired from multiple sensors when compared to other AI/ML algorithms.

Regarding the contribution to this area, the scientific contributions are already provided in the original version of the manuscript. In the original and revised version of the manuscript, scientific contributions are listed just after the hypotheses in subsection 1.3. However, in the revised version of the manuscript, the scientific contributions are rewritten to emphasize these contributions as much as possible. Citing from the revised version of the manuscript (before last paragraph in the Introduction section): 

The scientific contribution of this paper are: 

  • investigate the possibility of GPSC application to the publicly available dataset for the detection of fire, 
  • investigate if dataset balancing methods have any influence on classification accuracy of obtained symbolic expressions 
  • investigates if GPSC with random hyperparameter search method and 5-fold cross-validation can produce the symbolic expression with high classification accuracy in fire detection. 
  • investigate if using the best symbolic expression can produce high classification accuracy in fire detection on the original dataset.

  • Kindly provide key contributions of this research work in point-wise manner (may be at the end of Introduction section).  

Answer: The scientific contribution of this paper is already mentioned in the original version of the manuscript. Citing from the revised version of the manuscript (before last paragraph in the Introduction section): 

The scientific contribution of this paper are: 

  • investigate the possibility of GPSC application to the publicly available dataset for the detection of fire, 
  • investigate if dataset balancing methods have any influence on classification accuracy of obtained symbolic expressions 
  • investigates if GPSC with random hyperparameter search method and 5-fold cross-validation can produce the symbolic expression with high classification accuracy in fire detection. 
  • investigate if using the best symbolic expression can produce high classification accuracy in fire detection on the original dataset.
  • Why do you selected 5-fold cross-validation (i.e. k=5)?  

Generally, the 5-fold cross-validation is a commonly used method in Machine Learning. The benefit of selecting the 5-fold cross-validation is to ensure the trained model is robust and that overfitting did not occur. Overfitting is one of the common disadvantages of using a classic train-test process of ML methods. To prove that overtraining/overfitting of the model did not occur was to measure mean and standard deviation of evaluation metric values obtained on different dataset parts (e.g. train and test datasets). As seen from the results presented at Results and Discussion section small values of standard deviations of classification metric values were obtained (10^-4 - 10^-5).  The paragraph explaining the 5-fold cross-validation process was added at the beginning of the subsection entitled “5-fold cross-validation” in the revised version of the manuscript and is backed-up by additional literature. 

Citing from the revised version of the manuscript (first paragraph in the subsection “5-fold cross-validation of “Materials and Methods” section): “The 5-fold cross-validation process has been chosen to generate a robust symbolic expression that can be used to detect fire with high classification accuracy. The classical approach of dividing the dataset on train/test without the use of 5-fold cross-validation can produce high classification accuracy on a train dataset however, unseen data can result in poor classification accuracy. So classic train/test process can in some cases cause over-fitting. Some examples of 5-fold cross-validation applications can be seen in [47-49]. To prove that over-fitting did not occur the mean and standard deviation of evaluation metric values obtained on the train/test dataset were calculated. The high values of the standard deviation of any evaluation metric used in this paper could indicate a large difference between the evaluation metric values achieved on the train and test datasets, respectively. However, in this research, large standard deviation values of evaluation metrics used did not occur.

  • The authors are requested to add % improvement in the classification accuracy with the proposed method as compared to any other existing state of art, if possible.  

Answer: At the end of the results section a new subsection is provided entitled “Results Summary” in which the results are compared with the results of other investigations mentioned in the introduction section. This subsection also contains points from the Conclusions section that were suggested by the reviewer to be moved from the Conclusions section to the Results section (point g. of reviewer comments). 

  • The authors are requested to describe the software’s used to carry out this research work, and the computational complexity of the proposed algorithm, if possible.  

Answer: In the Computational Resources subsection of section “Materials and Methods” the authors described in detail what Computation Resources they used to conduct this research as well as the used software including all necessary libraries. Citing from the original and modified version of the manuscript: “In this paper, all investigations were conducted on a laptop with AMD Ryzen 5 Mobile 5500U 6-core (12 threads) processor, and 16 GB of DDR4 RAM. The Python programming language (version 3.9.1) was used to create all scripts. The original dataset was balanced using undersampling and oversampling functions from the imblearn library (version 0.9.1). The statistical analysis and correlation analysis was done using pandas library (version 1.0.5). The GPSC algorithm in these investigations was imported from gplearn library (version 0.4.1). The 5-fold cross-validation and random hyperparameter search method were developed from scratch. To visualize all the results and correlation heatmap the matplotlib library (version 3.4.3) was used. 

  • Please merge Section 3 (Results) and Section 4 (Discussion). It would improve paper readability further.  

Answer: In the modified version of the manuscript the Results and Discussion section were combined together in the section entitled “Results and Discussion” as requested by the reviewer.

  • Please try to reduce Conclusion section. You can move the points mentioned in it to result section. Also provide system limitations and future scope for this work. 

Answer: As suggested by the reviewer the points in the Conclusions section are moved to the end of the results section. In the conclusions section the concise form of this conclusions were given in one paragraph and after that the paragraph about advantages and disadvantages of used method are described as well as the guidelines for the future work. Citing from the revised version of the manuscript (“Conclusion” section): In this paper, the GPSC with random hyperparameter search method and 5-fold cross-validation was applied to publicly available datasets to obtain robust symbolic expressions which could detect fire with high classification accuracy. 

\color{cyan}The conducted investigation showed that GPSC can generate symbolic expressions which can be used for fire detection with high classification accuracy. The combination of GPSC with random hyperparameter search method and 5-fold cross-validation generated robust symbolic expression for fire detection with high classification accuracy. The dataset balancing methods ADASYN, SMOTE, and Borderline SMOTE balanced the original dataset and provided a good starting point for the application of the GPSC algorithm. Using these dataset variations to train GPSC produced symbolic expression with classification accuracy in the range of 0.97 up to 0.999. So, the dataset balancing methods have an influence on the classification performance of obtained symbolic expressions. The applied procedure also showed that the best symbolic expression in terms of classification accuracy obtained on a balanced dataset can achieve almost the same classification accuracy when applied to the original (imbalanced) dataset.  The investigation also showed that the best symbolic expression does not contain particular matter variables only temperature, humidity, TVOC, Raw H2, Raw Ethanol, and air pressure.

The advantages of the proposed method are: 

  • \item after application of GPSC the symbolic expression is obtained that can be easily used regardless of its size since it requires lower computational resources to produce the solution when compared to other ML algorithms,
  • \item the dataset balancing methods created a good starting point for the implementation of GPSC and using GPSC symbolic expressions with high classification performance were obtained,
  • \item the GPSC with random hyperparameter search method and 5-fold cross-validation generated the symbolic expressions that are robust and have high classification accuracy.

The disadvantages of the proposed method are: 

  • \item to implement a random hyperparameter search method the ranges of each hyperparameter have to be defined by initial testing of GPSC. The population size, maximum number of generations, tournament size, and parsimony coefficient have a great influence on the execution time. The bigger the size of the population, tournament size, and larger maximum number of generations the longer it will take the GPSC to execute. However, the parsimony coefficient is the most sensitive GPSC hyperparameter and the range should be carefully defined. If the value of the parsimony coefficient is too small it can result in a bloat phenomenon while a very large value can prevent the growth of the symbolic expression which will result in a small symbolic expression with poor classification performance. 
  • \item the dataset oversampling methods greatly influence the GPSC execution time since the dataset used to obtain symbolic expression using GPSC is much larger than the original one. 

This approach showed how using a simple GPSC algorithm and the data obtained from the sensor fusion system a robust symbolic expression can be obtained which can detect fire with high classification accuracy. The symbolic expression can be potentially integrated into the micro-controller system which controls the entire multisensor system to provide additional information if the fire is actually occurring in a multi-sensor environment. The other benefit of using symbolic expression when compared to other ML models is that this expression requires less computational power and memory than ML models so it can be easily integrated with microcontroller devices such as Arduino.\newline 

The future work will be focused on development of multi-sensor system and collecting data to obtain a balanced dataset. After reasonably large and balanced dataset is collected other ML methods will be utilized to obtain ML models which could detect fire with high classification accuracy. After different models are trained they will be implemented on micro-controllers to measure the time required to detect fire by conducting different fire scenarios.

  • As the proposed work deals with AI and ML based system, I suggest citing following recent papers.
  • S. R. Jondhale, Manish Sharma, R. Maheswar, Raed Shubair, Amruta Shelke, “Comparison of Neural Network Training Functions for RSSI based Indoor Localization Problem in WSN”, 
  • Handbook of Wireless Sensor Networks: Issues and Challenges in Current Scenario's, Part of the Advances in Intelligent Systems and Computing book series (AISC), vol. 1132, pp.112-133, Springer Nature, 2020.

Answer: The suggested literature was included in the manuscript. Citing from the modified version of the manuscript:The environmental issues can be a major challenge in achieving high measurement accuracy with wireless sensor network (WSN) and the main reason for this is the noise uncertainty. To overcome this problem in [21]  the artificial neural network (ANN) was applied for the received signal strength indicator based indoor target localization in WSN. In this investigation the authors have investigated the performance of 11 different ANN training functions and the results showed that all trainin functions shows higher Average Localization Error and the system is more consistent in providing better location estimates.

Reviewer 2 Report

The introduction part might be expanded on Symbolic Expressions for Fire Detection and also to include difficulties that arise in the context of previous work.

The use of issues in the existing system related to Sensor Fusion and Fire Detection and how the author proposes to overcome them should be included in the literature review approaches. Authors may refer some more recent works like

Optical solitons with nonlinear dispersion in parabolic law medium and three-component coupled nonlinear Schrödinger equation

IoT enabled HELMET to safeguard the health of mine workers

The manuscript has some typos and alignment issues, so authors should check the paper completely and avoid jargon words.

Authors can elaborate on the results section and also should show a comparison with respect to the accuracy, quality of prediction, and precision by considering the Symbolic Classifier.

A few figures seem to be blurred and the authors should take care of the clarity.

Author Response

The authors of this paper want to thank the reviewer for his time and effort in providing constructive comments and suggestions which significantly improved the quality of this manuscript. The authors do hope that the paper in this form is acceptable for publishing. 

The answer to comments and suggestions made by the second reviewer is listed below. 

The introduction part might be expanded on Symbolic Expressions for Fire Detection and also to include difficulties that arise in the context of previous work.

Answer: The authors extensively searched for such papers regarding the Symbolic Expressions for Fire Detection/Smoke Detection. To the best of our knowledge, the authors did not find any paper in which this method was described. Generally, genetic programming is not commonly used in scientific research when compared to other AI and ML methods.

The use of issues in the existing system related to Sensor Fusion and Fire Detection and how the author proposes to overcome them should be included in the literature review approaches. Authors may refer some more recent works like

  • Optical solitons with nonlinear dispersion in parabolic law medium and three-component coupled nonlinear Schrödinger equation

  • IoT enabled HELMET to safeguard the health of mine workers

Answer: The papers suggested by the reviewer were included in the last paragraph of the second subsection in the Introduction section of the revised version of the manuscript. The authors have also included some additional literature to address the sensor fusion topic of this manuscript. 

The manuscript has some typos and alignment issues, so authors should check the paper completely and avoid jargon words.

Answer: The manuscript was thoroughly searched and the authors hope that in the revised version the jargon words are omitted. 

Authors can elaborate on the results section and also should show a comparison with respect to the accuracy, quality of prediction, and precision by considering the Symbolic Classifier.

Answer: Due to the request of another reviewer the Results and Discussion sections were merged together and by doing so improved the readability of the Results section. The Results section in the revised version of the manuscript is renamed to “Results and Discussion”. The authors do hope that now the achieved results are explained in detail. Another subsection was created (entitled “Results Summary”) in which the achieved results with this approach are compared with the results of other research papers described in the Introduction section. In this subsection, a detailed comparison, the advantages, and disadvantages of this approach are given.

A few figures seem to be blurred and the authors should take care of the clarity.

Answer: The authors have taken a closer look at all figures that are shown in the manuscript and have found that Figures 1 and 4 have low dpi so the dpi was improved to 1000 dpi. In order to make sure that all figures are sharp the dpi in all Figures was increased to 1000 dpi. 

Round 2

Reviewer 1 Report

No comments.